# Medical Imaging Applications of Federated Learning

**DOI:** 10.3390/diagnostics13193140

**Published:** 2023-10-06

**Authors:** Sukhveer Singh Sandhu, Hamed Taheri Gorji, Pantea Tavakolian, Kouhyar Tavakolian, Alireza Akhbardeh

**Affiliations:** 1Biomedical Engineering Program, University of North Dakota, Grand Forks, ND 58202, USA; hamed.taherigorji@und.edu (H.T.G.); pantea.tavakolian@und.edu (P.T.); 2SafetySpect Inc., 4200 James Ray Dr., Grand Forks, ND 58202, USA

**Keywords:** federated learning, medical imaging, brain imaging, COVID-19, pancreas, skin disease, breast imaging, computer vision, artificial intelligence, differential privacy

## Abstract

Since its introduction in 2016, researchers have applied the idea of Federated Learning (FL) to several domains ranging from edge computing to banking. The technique’s inherent security benefits, privacy-preserving capabilities, ease of scalability, and ability to transcend data biases have motivated researchers to use this tool on healthcare datasets. While several reviews exist detailing FL and its applications, this review focuses solely on the different applications of FL to medical imaging datasets, grouping applications by diseases, modality, and/or part of the body. This Systematic Literature review was conducted by querying and consolidating results from ArXiv, IEEE Xplorer, and PubMed. Furthermore, we provide a detailed description of FL architecture, models, descriptions of the performance achieved by FL models, and how results compare with traditional Machine Learning (ML) models. Additionally, we discuss the security benefits, highlighting two primary forms of privacy-preserving techniques, including homomorphic encryption and differential privacy. Finally, we provide some background information and context regarding where the contributions lie. The background information is organized into the following categories: architecture/setup type, data-related topics, security, and learning types. While progress has been made within the field of FL and medical imaging, much room for improvement and understanding remains, with an emphasis on security and data issues remaining the primary concerns for researchers. Therefore, improvements are constantly pushing the field forward. Finally, we highlighted the challenges in deploying FL in medical imaging applications and provided recommendations for future directions.

## 1. Introduction

Federated Learning (FL) is a machine learning, specifically decentralized learning, technique that allows multiple entities to train an algorithm collaboratively without exchanging their respective local data [1]. McMahan et al., the group that coined the technique, describe the approach as a loose federation of devices coordinated by a central server that collectively learns from separate datasets by passing model parameters to each other rather than the raw training data. The process utilizes secure encryption and communication advances to transfer models to clients rather than datasets [2]. In contrast to traditional machine learning examples, where institutions must share and centralize their data, FL instead shares only the model parameters, allowing for a common, robust model without sharing data [3].

Since McMahan et al. proposed the concept in 2016, researchers have utilized FL in several domains, such as edge computing, blockchain, IOT, and others [1,4,5,6]. Recently, however, many developments have occurred within the healthcare domain. FL’s inherent security features and convenience, coupled with the strict regulations governing patient-related data, establish a situation where FL could significantly improve research endeavors within the healthcare space.

Although there have been significant advancements in the healthcare space regarding artificial intelligence (AI), progress, specifically generalizability, has been stifled due to the lack of quality data available to researchers [7]. Availability of high-quality labeled data, although an inherent problem for any domain, is challenging in the case of health-related data due to its sensitive nature and strict regulations. Local and national laws such as the California Consumer Privacy Act (CCPA) and Health Insurance Portability and Accountability Act (HIPAA) in the United States or the General Data Protection Rule (GDPR) in Europe limit the transfer of personal data across organizations and countries even when data is anonymized. The sensitive nature of the data also requires enhanced security measures that often lead to a reticence toward sharing. From these struggles, researchers began applying FL to healthcare datasets.

In the past years, the number of papers published within the field of FL has increased drastically. While most of these papers vary in terms of their focus, the healthcare field is one of particular importance. As previously mentioned, AI has begun to transform the medical field, particularly medical imaging, which has been the subject of several remarkable achievements. As a result, this paper focuses on providing a survey regarding how researchers apply FL in medical imaging so that it can be easily referenced and built upon rather than focusing on the technical rigor of FL.

## 2. Review Methodology

### 2.1. Research Questions

To provide a structured and extensive overview of the relevant FL papers, we posed the following research questions.

RQ1: How does federated learning differ from centralized learning when dealing with medical imaging applications?

RQ2: What are the different tasks/scenarios federated learning is used in for medical imaging applications?

### 2.2. Search Process

The literature included in this review came directly from the results of ArXiv, IEEE Xplorer and PubMed searches. Some articles came indirectly from the cited works sections of the papers included. This was done to create as comprehensive a list of works as possible.

### 2.3. Inclusion/Exclusion of Literature

The articles have been selected based on FL and/or medical imaging. The date exclusion criteria applied to the results were relevant articles before January 2023, except for two articles. No exclusion criteria were applied regarding the type of publications. The articles included primarily focused on FL or provided important background information.

## 3. Results

In this review, 7708 articles were screened, and 103 were cited, as well as three websites. The flow diagram of the search process for the literature review can be seen below in Figure 1. Additionally, Table 1 below summarizes the different search terms used and the articles that resulted. As an example, when using the keyword “federated learning” ArXiv, including the quotation marks to restrict results to that phrase, resulted in 3291 results, while PubMed (PM) resulted in 354 articles; additionally, IEEE Xplore’s search resulted in 4063. Additionally, 18 other articles were identified by looking at the reference sections of the selected articles. Lastly, three of the websites referenced contained background information or information related to the discussion section. While the initial search term included some results, many were excluded because of the scope of the review paper. The use of other terms, such as medical and healthcare, were used to include more relevant results.

For each subsection of the paper (brain, pancreas, breast, COVID-19), separate, smaller literature searches were conducted. These searches started off as general searches but were aimed at understanding how federated learning has evolved over time in these respective areas. Attempts were made to find all relevant articles solely by database searches.

## 4. Background

While this review focuses primarily on medical imaging, other reviews provide valuable insight into FL’s background, status, and remaining challenges. To name a few, Castiglioni et al. wrote a review that provides excellent context around FL, providing an explanation of how AI in medical imaging has evolved, as well as discussing the basics of FL [7]. Joshi et al. and Pfitzner et al. spent a significant amount of time discussing the fundamentals of FL and the different variations explored over the years and relating them to various works conducted within the healthcare sector [8,9]. Nguyen et al. briefly discussed the reviews published before but then discussed FL’s advantages, requirements, motivations, and applications in a medical context [10]. While several reviews remain broad regarding their subject matter, others, such as Kaissis et al., focused on security and privacy-preserving applications, providing context to those related works [11]. Moreover, several other reviews focused on a particular disease, such as [12], which focused on ocular diseases. However, Chowdhury et al. focused on how researchers apply FL to various cancer detection scenarios [2]. Rauniyar et al., like this paper, provide an overview of medical applications rather than focusing on technical rigor [5]. Table 2 briefly summarizes the various reviews referenced in this paper’s writing.

### 4.1. What Is FL?

As previously mentioned, FL is a decentralized learning technique that allows multiple entities to train an algorithm collaboratively without exchanging their respective local data [1]. Depending on which approach of FL is applied, the workflow can change significantly. Figure 2 below depicts a commonly used server-centric version of FL. The first step is for the central server to distribute the global model to the participating client devices. These individual clients (nodes) then train on their dataset updating, creating a local version of the model. The nodes then send their versions of the model back to the central server. Depending on the aggregation schemes, the central server reconciles the variations from the local models to update the global model. This entire process repeats itself multiple times until the training is complete [11].

Although the setup above is the most used, researchers have devised variations of the original setup to suit their needs [8]. Variations in the number of nodes, presence of a central server, type of data, aggregation style, and communication protocol are all characteristics that would shape the approach and, thus, workflow. Due to their flexibility while still maintaining data governance, federated learning approaches have the possibility of becoming the most widely used next-generation privacy-preserving technique, both in industry and medical AI applications [11]. The types of variations can be classified into a few overarching categories: those that are hardware-inspired, those that are data-inspired, and those that are security-inspired. It is worth noting that this paper uses these categories to establish some organization for the research; however, these categories intertwine heavily. The following paragraphs give context to the solutions researchers have proposed in a later part of the paper.

### 4.2. Setup/Architecture

Data analysis, especially within the medical field, requires many resources. These resources encompass physical hardware, including hard drives, computers, and internet connections, and expenses related to electricity, heat dissipation, and maintenance teams. Moreover, these devices must be correctly secured and compliant when dealing with sensitive patient health information. As a result, traditionally, smaller institutions did not contribute to scientific research as much as their larger research-oriented counterparts. Due to the decentralized nature of FL, the methodology has allowed smaller institutions to contribute to developing a global model. As a result, numerous contributions to the field have focused on enabling multi-institutional collaboration by developing more resilient methodologies compatible with diverse hardware configurations.

Two primary ways to set up FL systems are centralized and decentralized, as depicted in Figure 3 above. This terminology should not be confused with centralized vs. decentralized learning, which describes how data is stored. All FL is a form of decentralized learning. The choice of architecture must consider the task at hand and how much any institution can afford hardware, communication, and time. The original proposition posed by McMahan et al. included using smaller cell phone-like devices in an edge-computing fashion to conduct FL methods [1]. However, due to security and cost-based reasons, most medical research institutions usually have a single computer that interfaces with the rest of the federation.

In centralized setups, a single central server is responsible for initializing a learning sequence, coordinating with participating trusted client devices, and updating the model. This system, although the most popular due to the ease of setup and low client numbers due to the nature of medical institutions, has some drawbacks that have led researchers to explore other options [28]. The major bottleneck with this setup is that client devices usually communicate only with the server; hence, the server acts as a system bottleneck. A second drawback with centralized settings is that it provides the entire system with single-point failure, so the entire system is affected if there is some sort of issue with the central server. In contrast with the centralized setup, in a decentralized setup, the client devices can communicate to train a global model and update it directly with each other without any central server. A decentralized setup addresses and thus prevents the single-point failure issue but introduces a higher complexity level due to the infrastructure required. Researchers have identified device heterogeneity as another crucial hardware factor in federated learning, resulting from variations in computer hardware speed and connection speed [29]. For more information regarding the various types of FL setups concerning the computational burden, as discussed in this section, please refer to the work of Beltran et al. [16].

### 4.3. Security

Once this network of computing devices is identified and created, the next series of issues arise when figuring out how they will communicate. Due to their geographic locations and device type, it is improbable that a dedicated offline setup is achievable. FL infrastructure offers an approach to privacy and data, but there is a need for further implementation of measures to expand its privacy-preserving goals. Therefore, it is no surprise that the second most prominent research topic within FL is security.

Researchers have taken two main ways to improve security: anonymization/deidentification/pseudonymization and encryption [11]. Deidentification is removing any information that may lead to identifying the patient. In the case of HIPAA, there are 18 types of identifiers that, if removed, can transform PHI into anonymized data. Pseudonymization, on the other hand, is replacing certain identifiable parts of data, such as name, with a pseudonym or a fake piece of information (e.g., Samuel Clemen’s pseudonym was Mark Twain). However, these methodologies become complex depending on the type of imaging. For example, researchers have demonstrated that the capability of reconstructing the contours of a patient’s face from a computed tomography (CT) scan of a head facilitates reidentification, whereas tracing back an X-ray of the leg to an individual is more complicated. Encryption, however, uses cryptography on data so that any party that intercepts data would have a more difficult understanding. Studies have demonstrated that data can leak or tamper with algorithms, highlighting the importance of encrypting communication. Inherently, neural networks represent a form of memory; therefore, it is possible to reconstruct training data solely from the model’s weights alone. Model inversion or reconstruction attacks refer to reconstructing images from the model weights, resulting in unacceptable data leakage [11].

As a result, researchers often employ two primary forms of privacy-preserving techniques that implement the two methods discussed above: differential privacy (DP) and homomorphic encryption (HE) [11,30,31]. DP is the approach of reducing recognizable information about an individual while still maintaining the global statistical distribution of a dataset, e.g., researchers can determine the relationship between body mass index (BMI) and insulin levels without knowing the individual patient’s BMI, thereby preserving privacy. The implementation DP ranges from simply shuffling the input data to more commonly introducing noise. Applying DP can involve applying the input data, the computation results, or the algorithm. A significant drawback of DP is that data manipulation often results in degradation, which can be problematic when data is scarce.

Moreover, DP can easily be applied to tabular data but is more complex in medical imaging. Homomorphic encryption (HE) is regarded as the gold standard by information systems, allowing for certain mathematical operations on encrypted data as if unencrypted. The main trade-off with HE is efficiency, specifically computational performance.

Security is an important topic within the field of FL; the lack of a secure physical connection initializes the need for security. The implementation’s basis and nature depend on the data being analyzed or communicated. Data composition, distribution, and reconciliation are the next overarching theme of FL research topics. For a summary of the different kinds of attacks and more information regarding the security-based implementations, please reference the following reviews [9,11,32,33]

### 4.4. Learning Schemes

Another way data characteristics shape FL approaches that we will cover, but by no means is the only remaining way possible, is how data determines the learning type for FL. FL can be broken down into three main task categories: Horizontal, Vertical, and Transfer. These follow the same core principles of the FL paradigm but are distinguished by how their data sources differ.

In the case of Horizontal learning, each site has a different set of users in their data, but all the users share a similar feature set as depicted in Figure 4a. For instance, four hospitals or institutions in different countries collect MRI data on Alzheimer’s disease (AD) in the same feature space. Given that each hospital trains the model based on its data using the Horizontal FL, these four hospitals can develop the training model collaboratively, increasing sample size and, as a result, the model’s performance, reliability, and generalizability.

Vertical FL, also named feature-based FL, is when multiple sets of data with different features (feature space) from the same sample can be combined to help create a single decision [9]. Vertical FL can increase the feature space dimension by combining different features as depicted in Figure 4b. For example, three different hospitals or institutions in a city perform cognitive tests, MRIs, or blood gene expression on AD patients. Many patients probably visit these institutions since these three tests are the most common AD testing. So, there will be a significant intersection of patients, and Vertical FL can aggregate these different tests as features for the same patients to increase the feature space for more robust and reliable training of ML models. In medical imaging, it would be possible for a patient with both a CT and an MRI of the same area to be combined and the insights of both to be combined to make a singular decision [34].

The third method of FL is Transfer learning. Transfer learning can be used where both feature space and sample space of the clients’ datasets have a relatively small overlap as depicted in Figure 4c [9]. Following the above example for diagnosis of AD, consider three institutions or hospitals in three different countries, each of which does one of the cognitive tests, MRI, or blood gene expression on the AD patients. Because of the geographical distance, the sample space might have no overlap or barely overlap of information. Moreover, each institution or hospital does a different test on AD patients, resulting in no overlap in feature space. In such scenarios, Transfer learning can create an efficient and reliable model while transferring and adapting knowledge from different but related tasks. A subcategory of Transfer learning is called domain shift, which occurs when there are significant differences between institutions and their practices, ultimately causing a change in the data distribution between the algorithm’s training set and the dataset it encounters once installed [35]. Another example of Transfer learning in medical imaging would be for the task of using a model that can identify a pancreas in one set of imaging to conduct tumor segmentation of the brain in another set of imaging [36].

Advances in ML within the medical imaging field have been greatly afflicted due to the sensitive nature and variety of medical data. Therefore, FL has been applied in different ways to help compensate for the various shortcomings of traditional methods.

### 4.5. Data Partitioning

The largest and most prominent set of topics in FL is data-centric. The quality, type, and distribution of data significantly impact the approach one must adopt. Furthermore, the data available in medical imaging datasets are scarce and specific, creating a unique environment for researchers to explore.

As previously mentioned, an ML model’s quality depends on the data quality. ML aims to conduct tasks that generally require significant amounts of time, attention, and training. Creating a large dataset is not as simple as simply aggregating the data in a single location and randomly picking images to create a training set. Instead, curating the training data and understanding how to train a model properly is essential to achieve the best possible result. A model requires training on a diverse range of cases or a substantial volume of heterogeneous data to achieve generalizability.

A second characteristic of data sets is how well distributed the data is amongst the categories that make it up. In ML, specifically, to properly evaluate the performance of a model, the data must adhere to the assumption that it is Independent and Identically distributed (IID). For data to be IID, each random variable has the same probability distribution as the others, and all are mutually independent. However, in the case of real-world data, the distribution is rarely IID and thus must be accounted for by researchers.

These two characteristics plague the world of medical imaging. Data from a single source can be significantly biased based on the equipment, demographics, protocols, and pathologists labeling the data [2]. Researchers have shown that training data only from a single source tends to skew the performance of that population. In the case of medical imaging, specifically histopathology images, deep models tend to fail to generalize well when used in a different hospital. The diverse imaging methods, devices, and types of annotations would also cause issues with the IID premise [8,37,38]. FL, however, set out to help alleviate non-IID issues, overcoming the biases of a particular institution by aggregating with the models of other institutions [22].

### 4.6. Aggregation Methods

Another data-based research topic within FL is regarding how the global model updates or the aggregation technique. Step 4 in Figure 4 depicts when the aggregation techniques are used within the FL process. The methodology of reconciliation of a model and architecture can influence the resulting global model’s accuracy. The most used aggregation method is FedAVG; in this technique, the global model is simply an average of the local models [8]. Aggregation techniques help to optimize non-IID data distributions and imbalanced properties of the data. However, as shown in Figure 3, variances in data distributions, data biases, and limited resources have led researchers to explore other techniques [39]. 

## 5. Medical Imaging Applications

FL was implemented to help facilitate collaboration between groups and institutions; however, this has led to a series of issues requiring researchers to develop new tools to address them. The information and background provided in the previous section will help provide some context for the applications we will mention. In this systematic review, our objective was to explore the current state of FL in medical imaging by exploring the different workflows, architectures, algorithms, and frameworks deployed by other groups that have implemented their applications. In this review, we will categorize the author’s contributions based on the part of the body being imaged, the disease, and the task of machine learning. A high-level overview of the categories is depicted in Figure 5 below.

### 5.1. Brain

The need for data sharing in neuroimaging was evident to researchers early on. Thus, the field embraced data collaboration before other fields. Radiographs of the head are much easier to link back to patients due to the contours of the face present in them. Therefore, the need for security on these images was more significant than on the image of a foot, for example [11]. As a result, public funding agencies and journal publishers have set forth mandates that ultimately resulted in the development of data repositories and consortia [40]. Additionally, efforts such as the Collaborative Informatics and Neuroimaging Suite Toolkit for Anonymous Computation (COINSTAC), as well as other analysis frameworks, were also established within the field (for a more comprehensive list of neuroimaging platforms and tools, please refer to [23]). The efforts of these research groups are organized by the disease they applied their FL efforts towards; the main ones include tumor detection, Alzheimer’s/Parkinson’s detection, general structure classification, and other disease groups. Table 2 lists and summarizes efforts within FL that relate to the brain.

#### 5.1.1. Brain Tumor Detection

The first application of FL on medical imaging data was conducted on a brain MRI dataset by Sheller et al. in 2018 [41]. The first of two publications by the group was conducted as a case study that implemented semantic segmentation on the public BraTS dataset. The group compared their FL efforts to two other alternative collaborative learning techniques (institutional incremental learning and cyclic institutional incremental learning) [41,42]. Li et al. published their work shortly after Sheller et al.’s first work [43]. This group focused its efforts on security by implementing a parameter-sharing method that would ultimately combat model inversion techniques that can be used to reconstruct the training examples. The group also implemented gradient clipping as a model regulator to prevent over-fitting using the same BraTS dataset as Sheller et al.

As FL’s popularity increased, variations in how it was implemented began to occur. Fay et al. was a group that focused on a different type of FL called Private Aggregation of Teacher Ensembles, which trains local models and then labels unlabeled data that then makes up the global model [44]. In late 2021, a group, Machler et al., also won a challenge that utilized the Federated Tumor Segmentation Challenge 2021 [45]. The group implemented a new method called FedCostWAvg, which aggregated the models trained on different data differently than the gold standard of FedAVG to ultimately be the best-performing brain tumor segmentation algorithm. In August of 2021, Knolle et al. introduced MoNet, an FL architecture that reduces the number of parameters needed for FL, thus allowing it to be used in resource-constricted environments [36]. The group used CT images from the Medical Segmentation Decathlon to conduct their experiments. Later, in 2021, another group, Ads et al., implemented a nontraditional FL model by being the first split learning and Vertical distribution FL for brain tumor classification [34]. Split learning differs from the traditional FL methods by splitting how the training is done between the client and server.

In 2022, He et al. worked on how information is shuttled back and forth/communication [46]. As model complexity grows, the cost of communication in terms becomes a significant bottleneck for the FL system. As a result, the group conducted a case study that implemented a cosine-based quantization scheme that encrypted and compressed the model weights and gradients, allowing for a security-based improvement to FL. Around the same time as He et al. [46], Zhang et al. [29] created a method that leveraged network level split and feature map concatenation strategy to help combat statistical data heterogeneity, an issue that afflicts FL models. This group implemented their case study on the BraTS dataset to conduct tumor segmentation and on simulated data from retinopathy and bone age data. Additionally, Rawat et al. introduced a Robust Learning Protocol when participating in the Federated Tumor Segmentation Challenge in 2022 [47]. Through their method, the group combined server-side adaptive optimization and parameter (weights) aggregation schemes to address data heterogeneity issues and the communication cost of training.

Later, in 2022, Islam et al. claimed to be the first to apply a complex convolutional neural network (CNN) model to FL [48]. The group locally trained a complex CNN model by combining the weights of DenseNet21, VGG19, and InceptionV3 models to create an average CNN model. They then implemented this average CNN model in an FL setup and evaluated its performance. They found that the resulting model performed well compared to the other models; however, not as well as a locally trained average CNN model. The group concludes that although the performance of the FL average model is slightly less than the locally trained average model, it is worth the privacy-preserving benefits that the FL model offers.

In 2022, Pati et al. presented the most extensive FL study spanning 71 different sites to detect glioblastoma sub-compartment boundaries [49]. This group did what other groups aspire to one day conduct: an at-scale FL model. Pati et al. presented many insights. However, the consensus was that FL at scale was, in fact, more effective than locally trained models.

#### 5.1.2. Alzheimer’s/Parkinson’s

Shortly after the contributions of Sheller et al. in 2020, Silva et al. wrote a series of papers that proposed an open-source framework for federated learning in healthcare [50,51]. The group demonstrated their framework by creating a model that analyzes subcortical volumes and cortical thicknesses through MRIs to help identify the most critical dimensions to help classify healthy Alzheimer’s and Parkinson’s imaging.

In 2021, Huang et al. noticed that data, when pooled together from multiple institutions, especially MRI data, is susceptible due to the variability of scanners and sites [52]. This variability is due to the acquisition protocols, recruitment criteria, and different machines, not to mention the variability from the labeling. As a result, the group proposed a framework named Federated Conditional Mutual Learning or FedCM and applied their framework on T1w MRIs to be the first federated learning on multi-dataset Alzheimer’s disease classification by 3DCNN.

In 2022, Stripelis et al. utilized an FL framework they proposed in their previous works and applied it to MRI datasets to classify Alzheimer’s disease and estimate Brain Age [53]. While the group’s previous works detailed their security enhancements and architecture development, this later work was more of a case study implementing their previous works on a heterogeneous dataset to prove the capabilities [31,39,53].

In late 2022, Dipro et al. applied FL to Parkinson’s Disease detection in Single-photon emission computed tomography [54]. This group utilized data from the Parkinson’s Progression Markers Initiative to conduct image classification to detect Parkinson’s Disease.

#### 5.1.3. General Brain Structure Classification

In 2019, Roy et al. proposed and created an FL framework that looked to cut out the need for a central server, allowing for another level of decentralization and, hence, more security [28]. The group implemented their framework on T1 MRI scans and carried out whole brain segmentation. The group named this framework BrainTorrent.

In 2021, Bercea et al. proposed another framework called federated Disentanglement or FedDIS to help combat issues of data heterogeneity [55]. The group found that when analyzing MRI images of the brain, the anatomical structures remain similar across institutions. Therefore, sharing only shape characteristics of abnormal structures with clients would be more beneficial. The group conducted their experiments on data from multiple sites and found they could outperform the state-of-the-art auto-encoder by 42%.

Later, in 2021, Parekh et al. demonstrated a cross-domain application by demonstrating the ability to transfer between PET and CT scans [56]. The group was able to demonstrate a cross-task model by applying a model trained on brain lesion segmentation and transferring it to breast lesions in multi-parameter MRIs.

Brain Age is the estimation of the person’s age from their brain structural MRI scan. A difference between the person’s actual age and the predicted age has been proven to be a valuable biomarker for the early detection of various diseases. As a result, the works of Stripelis et al. and others have utilized this biomarker [31,39,53]. Moreover, Gupta et al. sought to demonstrate vulnerability within FL learning setups by demonstrating the ability to conduct an attack known as a membership inference attack successfully [57]. The group’s work was used to complement those of Stripelis et al.

#### 5.1.4. Others

##### Dementia

The later work of Stripelis et al. sought to improve security by proposing a framework that utilized homomorphic encryption to alter how the central model is updated [31,39]. They applied their framework to MRI scans that would help determine the age of the brain by analyzing the structures present, ultimately classifying stages of Dementia rather than Alzheimer’s in their previous works. Shortly after their initial work, Stripelis et al. worked on an architecture called MetisFL that encrypts the parameters before transmission, computes the community model under fully homomorphic encryption, and uses information-theoretic methods to limit leakage [53].

##### Autism

During the time of Stripelis et al., another group, X. Li et al., was trying to implement a privacy-preserving strategy to be used in multi-site fMRI classification [35]. Unlike the other groups mentioned before, this group was using fMRIs instead of MRIs to experiment with identifying robust biomarkers for Autism Spectrum Disorders (ASD) on the Autism Brain Imaging Data Exchange (ABIDE) dataset. The group proposed an FL approach where random, Gaussian and Laplace mechanisms alter the model weights. The group also claims to be the first group to investigate domain adaptation, which is when data issues arise due to utilizing medical images from different institutions. Another group that utilized the ABIDE dataset was Fan et al.; this group set out with two primary goals and they were the first to create a federated learning framework for 3D medical images, specifically for multi-site 3D brain MRI images [58]. Due to the nature of the data, they also implemented privacy security measures to keep their data secure. The group set out to use their framework to accurately diagnose autism spectrum disorder, much like X. Li et al.

Shamseddine et al., in 2022, conducted a series of experiments with an FL framework where they predicted whether a patient was diagnosed with ASD based on two methodologies [59]. The first methodology was based on a behavioral screening where the responses were analyzed, and then the diagnosis of ASD was rendered. The second experiment was conducted using a clear facial image of patients.

##### Multiple Sclerosis

Liu et al. is a group that worked on Multiple Sclerosis lesion segmentation and found that the way the disease presented itself in a scan and domain shifts within datasets left current FL techniques subpar when set for this task [60]. The group then set out to create a new framework to modify the weight placed on parts of their training data based on the local node and volume of the lesion. This framework, coined as FedMSRW, claims to be the first of its kind, outperforming other FL methods.

##### Brain Metastasis

Huang et al. identified and implemented a way to implement continual learning for the difficult task of brain metastasis identification [61]. The group utilized the DeepMedic neural network and achieved identical results to mixed data.

##### Schizophrenia and Depressive Disorders

Another disease state susceptible to domain shift issues is using fMRIs to diagnose Schizophrenia and major depressive disorders. Zeng et al. set out to create a method known as GM-FedDA, where a two-stage method can be implemented to increase performance [62]. The group showed that better performance can be achieved by using one common source adversarial domain adaptation strategy and fine-tuning the model using a gradient matching method. The group demonstrated the ability of this method on Resting-state functional MRIs for diagnostic classification of Schizophrenia and major depressive disorder.

##### MRI Reconstruction

MRIs have transformed the world of medicine by allowing clinicians to picture the inside of the body non-invasively. MRIs are incredibly complicated; one aspect that has been shown to increase efficiency is the reconstruction algorithms. Deep learning models have been developed to help in this endeavor; however, they often fall short because they typically show poor generalization. Therefore, in 2021, Guo et al. proposed a framework called FL-MR that would enable FL for MRI reconstruction [63]. Additionally, the group identified that domain shift is a significant issue in this application and thus proposed to align the latent space distribution between the source and target domains. Moreover, in 2022, Elmas et al. worked on creating an MRI reconstruction model that utilized a two-stage FL-based approach [64]. The group approach included across-site learning of a generative MRI prior and prior adaptation following injection of the imaging operator. A note regarding these two tasks: while these papers focus on medical MRIs, they do not, however, diagnose a particular disease; therefore, in Table 3, they have been labeled as Non-Diagnosis.

### 5.2. Chest and Abdomen

#### 5.2.1. COVID-19

Over the past few years, there have been monumental strides when it comes to Deep Learning in Chest x-ray radiology. The influx of annotated chest X-rays has spurred significant contributions due to the recent COVID-19 pandemic. The COVID-19 pandemic that struck the world in 2020 tested the world in many ways; one of the most prominent needs throughout the pandemic was the need to share data and knowledge with colleagues and institutions worldwide. FL was a prime candidate in the minds of researchers at the time to help solve issues and make a better set of machine learning models. As a result, one of the most common applications of FL in medical imaging is regarding COVID-19. A summary of COVID-19 applications of FL can be seen in Table 4.Unlike the previous Brain section, which had several diseases classified under that part of the body, COVID-related studies all relate to the lungs; therefore, we will be splitting the sections based on the modality utilized.

##### COVID-19 Chest X-rays

The first application of FL to COVID-19 was conducted in July 2020 by Liu et al., who focused on comparing the performance of FL and non-FL models [65]. In total, Liu et al. compared the performance of four different ML models on CXR images from the publicly available COVID-19 x database. Lydia et al. is another group that used the COVID-19 x database to implement FL. The group focused on creating a detection model on an IOT-enabled edge computing environment [68].

In 2021, Feki et al. also implemented FL on COVID-19 detection in Chest X-rays. Their work also showed that FL can perform well on datasets with skewed distributions and conducted several experiments comparing the decentralized and centralized models [72].

Longling Zhang et al. focused on enhancing the security of FL by implanting a Generative Adversarial networks framework [30]. The group also demonstrated through their work that they could alleviate issues in non-IID data.

##### COVID-19 CXR + EMR

Salam et al. built upon the works of Liu et al. and 33 others by creating a custom ML model and optimizing FL parameters [73]. Additionally, Salam et al. added descriptive data to their X-ray images. The group studied the efficacy of FL vs. traditional learning by developing two ML models based on Keras and TensorFlow; the group also tried to identify which parameters affect model prediction accuracy. Dayan et al., another group, significantly highlighted the differences between FL and non-FL models. Dayan et al. had previously published work within the field using local ML models and transferred their knowledge into their FL findings [69]. The group found that their approach and use of FL allowed their model to become more generalizable, overcoming certain data biases. Dayan et al., much like Salam et al., had multidimensional data that included EMR and Chest X-ray images. Data quality and robustness were another primary focus among groups implementing FL in COVID-19 imaging applications.

Another group, Ho et al., found that using non-IID or similar-looking data is the source of a significant issue in FL systems [76]. The group went on to find that the size and distribution of the data sources can directly affect the performance and thus were able to increase performance by increasing the total number of clients, parallelism (client–fraction), and computation per client (batch size). Finally, Ho et al. found that splitting the dataset not only provided better results on the training of the model but also helped to increase the security of the data.

##### COVID-19 CT

One of the first applications of FL to COVID-19 CTs was done by Xu et al. This group identified a need for a large, diverse, and generalized dataset and thus set out to create a collaborative in which researchers could share data [66]. In 2021, another group, Kumar et al., also applied FL to CTs; however, they also proposed a way to normalize the data, proposed a novel COVID-19 detection technique, enhanced security through blockchain technology, and introduced a new dataset [75].

Lydia et al. focused heavily on the Internet of Things and edge computing-enabled computing environments. They used the publicly available COVID-19 x dataset to develop a detection model. The model utilized the Squeeze Net model, and the parameters were optimized with glowworm swarm optimization. The group ultimately classified images as normal, pneumonia, or COVID-19. They then compared their findings to other methods and found that FL COVID-19 techniques outperformed the other methods [68].

Shortly after, Salam et al. built upon the works of Liu et al. and 33 others by creating a custom ML model and optimizing FL parameters [73]. Additionally, Salam et al. added descriptive data to their X-ray images. Salam et al. also concluded that the federated machine learning model performs better in terms of accuracy and loss; however, the time required by this method is longer than that of traditional machine learning models [73]. Dayan et al., another group, significantly highlighted the differences between FL and non-FL models. Dayan et al. had previously published work within the field using local ML models and transferred their knowledge into their FL findings [69]. The group found that their approach and use of FL allowed their model to become more generalizable, overcoming certain data biases. Dayan et al., much like Salam et al., had multidimensional data that included EMR and Chest X-ray images. Data quality and robustness were another primary focus among groups implementing FL in COVID-19 imaging applications.

In a multinational study, another group, Dou et al., focused on demonstrating the feasibility of using FL for detecting COVID-19-related CT abnormalities. The group used datasets from four multinational centers to show the benefits of FL. The group also did a longitudinal case study estimating lesion burden at these different institutions [71].

Yang et al. looked to create a robust and generalized dataset by consolidating the data from numerous geographical locations using FL to conduct semi-supervised learning and label previously unlabeled data to define areas of interest within 3D chest CTs [37]. Ultimately, the group focused heavily on the issues around domain shift and demonstrated the utility of FL.

Durga et al. proposed a novel framework that was based on blockchain and FL [78]. The group utilized lung CT images from multiple publicly available datasets and compared the performance of their proposed model with the other existing model architectures in predicting COVID-19 while effectively preserving privacy.

##### COVID-19 CT + Clincal Data

Liang et al. worked on a framework that utilized CT imaging with and without EMR data to detect and distinguish COVID-19 from other lung issues and also worked on a model that could automatically segment the lesion within the CT, tracking the progress of disease over time [75].

##### COVID-19 CT/X-rays

Zhang et al. ran their models on both CT images and X-rays. The group found that rather than constantly updating the model, it should only be updated when the new data improves the model’s performance, proposing a dynamic fusion-based approach [70]. This scheme not only helps battle the issue of homogeneous data but also helps to save on the cost of computation and transmission of data but only updates when necessary.

##### COVID-19 X-ray and Ultrasound

Qayyum et al. proposed a clustered federated learning framework that processed multi-modal imaging data [77]. The group utilized X-ray and ultrasound imaging. The group compared CFL with conventional FL.

#### 5.2.2. General Chest X-rays

Ziegler et al. evaluated the feasibility of differential private FL on chest X-ray classification [32]. The group focused on the security aspect of FL by introducing Reyni differential privacy with Gaussian noise into the local training model.

### 5.3. Pancreas

Pancreatic cancer is the second leading cause of cancer-related death in American males [80]. Over the years, machine learning has made significant strides within the field. However, there have not been many instances where research groups have implemented FL in pancreatic tumor segmentation. A summary of pancreas related applications of FL can be seen in Table 5.

Wang et al. were the first to perform FL on pancreas segmentation data hosted at multinational sites [81]. The group used a set of abdominal CTs to conduct pancreas segmentation to highlight tumors and the pancreas itself. Shen et al. collaborated with Wang et al. to create one of the first pancreas segmentation to highlight the pancreas and the tumors [82]. This paper differs from Wang et al. because they use a 3rd publicly annotated dataset. Additionally, this group aimed to investigate three main points, introduce dynamic task prioritization for each task in multi-task learning, investigate dynamic weight averaging aggregation, and compare the effects of FEDavg and FEDprox on pancreas segmentation tasks—a multi-task FL method. The group’s experiments found that FL Dynamic Weight averaging model aggregation performed best but required significant manual tuning. Overall, the group focused its efforts on optimization and how the models are updated. Knolle et al., as mentioned in the brain section, created an efficient FL architecture that minimized the number of parameters required to conduct semantic segmentation on not only brain CTs but also pancreatic CTs [36].

### 5.4. Breast

Breast cancer is the leading cause of cancer-related deaths in women; as a result, women are screened for breast cancer regularly. Due to an 11% positivity rate, the datasets at these institutions tend to be overwhelmingly filled with negative results. As a result, traditional machine learning becomes difficult, with the dataset skewed in a particular direction. Researchers have combined datasets from other institutions to make a better classification system, providing a case for FL within the task. A summary of breast cancer applications of FL can be seen in Table 6.

The first implementation of FL in breast cancer classification was done by Roth et al., who applied FL to BIRADS, which was applied to a series of multi-institutional mammography data images [83]. These images are 2D X-ray images focused mainly on breast density calculation. Shortly after Sanchez et al., the group built upon the works of Roth et al. by continuing their pursuit of implementing federated learning in breast cancer classification [84]. The group implemented the first use of Curriculum Learning to boost classification performance while improving domain alignment. Lastly, Agbley et al. aimed to apply federated learning to detect breast cancer and appropriately classify them into subtypes [85]. This data differs from the predecessors because it uses local Invasive carcinoma of no particular type (IC-NST) from the breast histopathology image dataset. The group implemented FL on this dataset and experimented with a second set of neural networks that implemented Gabor kernels to extract another set of features.

**Table 6 diagnostics-13-03140-t006:** Applications of FL to Breast Cancer.

Author	Task	Goal
Agbley et al. [85]	Tumor Segmentation	Leverage FL to securely train mathematical models over multiple clients with local no special type images from the BIH dataset.
Roth et al. [83]	Breast Density classification	Create an FL model that can classify breast densities using BI-RAD data
Sanchez et al. [84]	Breast Cancer Classification	Create a novel memory-aware curriculum learning method for FL.

### 5.5. Skin

Skin diseases have increased in prevalence dramatically in recent decades. Variations in skin, disease, and tumors, as well as different resolutions, complex contexts, privacy concerns, and sensitive body part images, make it challenging to create an ML algorithm that is generalized enough to perform well. The datasets available often limit these efforts, making it a prime candidate for FL. A summary of skin disease related applications of FL can be seen in Table 7.

Hashmani et al. proposed an FL-based skin disease model that consisted of efforts to diagnose the type of disease [86]. Mou et al. created an FL model that conducted statistical and image analysis on skin lesion data across three Germany-wide stations [87]. Hossen et al. implemented their custom dataset to classify if a picture was one of four skin diseases [88]. The group found that their FL model showed less accuracy than CNN algorithms but also noted that the accuracy will increase daily as the number of training images increases.

Wicaksana et al. proposed Customized FL (CusFL) and demonstrated its ability to detect prostate cancer and identify skin cancer [89]. This group’s approach differed from traditional FL by iteratively training a client-specific model based on the global model instead of training a single one, thereby avoiding catastrophic forgetting.

### 5.6. Prostate

Prostate cancer (PCa) is worldwide the second most common cancer. Moreover, it is ranked fifth in mortality among men regarding cancer-related deaths [90]. Imaging plays a pivotal role in the staging process; International guidelines suggest using mpMRI’s PET or CT with Protein Specific Membrane Antigens. While there have not been many groups that have explored PCa, some have tried. A summary of prostate cancer applications of FL can be seen in Table 8.

In 2020, Yan et al. proposed a variation-aware federated learning framework to minimize client variations by transforming images into a common image space [91]. The group tested their framework on prostate cancer datasets intending to classify images. Additionally, Sarma et al. conducted a case study demonstrating their ability to utilize FL to train a deep learning model across three academic institutions while preserving patients’ privacy [92]. Furthermore, Wicaksana et al., as mentioned in the skin section, proposed a new way to train the FL model and demonstrated the abilities of the said model on prostate cancer data [89].

### 5.7. Others

FL has been applied to several other fields, and it is growing by the day; however, due to the number within each discipline, we decided to simply mention them in this section. Some of these other fields include ophthalmology [12,93,94,95], cardiac [96,97,98,99], larynx cancer [100,101], thyroid [102], and tuberculosis [103].

A noteworthy variation was conducted by Kassem et al., who used surgical videos rather than images to apply FL for surgical phase detection (not diagnosis) on Cholecystecotomy procedures [104].

## 6. Discussion

### 6.1. Research Questions

RQ1: How does federated learning differ from centralized learning when dealing with medical imaging applications?

Centralized learning is the traditional way ML is conducted, where data is pooled together in a single location, usually locally, for medical imaging applications. Constructing such a database requires extensive calibration and exhaustive compliance regulation with HIPAA practices. The financial, time, and risk costs associated with creating such datasets have led to only a handful existing, ultimately stifling progress in ML in medical imaging.

FL, however, was created to help address the limits of centralized learning. FL boasts the ability to train a model collaboratively without requiring all the clients to have access to each other’s data while still reaping the benefits of a model that has seen all the data. Moreover, FL can overcome some issues posed by non-IID datasets and, in some cases, combat the catastrophic forgetfulness of models by altering the importance of a client’s data. These benefits are significant when dealing specifically with medical imaging applications.

ii.RQ2: What are the different tasks/scenarios federated learning is used in for medical imaging applications?

COVID-19, brain, breast, and pancreatic are some of the most common implementations of Federated Learning in the medical imaging space. The main categories in which most research groups tend to focus their efforts are a proof of concept, comparing centralized learning to decentralized (federated learning), enhancing the security of federated learning applications, and finding ways to compensate for the fact that real-world data tends to be non-IID.

The application of FL to brain imaging is one of the most advanced partially since it is where Sheller et al. started off their applications. COVID-19 applications are also quite involved and have been quickly developing due to the onslaught of the pandemic and the need to share findings quickly and efficiently. The other applications, in comparison, remain few and far between; however, they have an overall positive trajectory.

### 6.2. Future Direction

Since McMahan et al. proposed the idea of Federated Learning, it has gained popularity among various fields that involve the use of machine learning and artificial intelligence. As a result, the number of research articles and applications of FL has been increasing in an exponential fashion as depicted in Figure 6. In the field of medical imaging, the application of the method to specific datasets remains largely unexplored; therefore, there are numerous articles that are written with the sole purpose of documenting the application of FL to a particular task/modality. Many of the applications within this category encompass semantic segmentation and identification tasks. Moreover, as the background section mentions, there is also a significant amount of research into enhancing the different features of FL, such as security and communication protocols. Another area of further research will be in the use of videos rather than images [104].

The future of FL in medical imaging will continue to increase; there will be more articles documenting the application to new datasets. Additionally, the focus on security and, consequently, advancements will be documented and required as FL becomes more widely implemented. Although FL makes collaboration easier, the acquisition of partners to implement any model still remains difficult; therefore, the utilization of Transfer learning, taking the models trained on similar modalities or tasks, will be used to enhance tasks. The increased levels of interest within the field have resulted not only in research interest but also piqued the interest of corporate institutions.

## 7. Conclusions

The application of FL to medical imaging aims to address issues that previously plagued the community. However, just as medicine is moving towards a personalized manner, FL must also be adapted to the specific needs of the image or disease state. Certain data types require more heterogeneous data; some require more security, while others are readily generalizable. Moreover, while security-wise, FL is inherently more secure than the outright sharing of data between institutions, model performance remains comparable to centralized learning counterparts. As advancements in understanding how ML models learn, perform, and can be transferred, it seems that it is only a matter of time before FL models will regularly surpass centralized models in most categories.

## Figures and Tables

**Figure 1 diagnostics-13-03140-f001:**
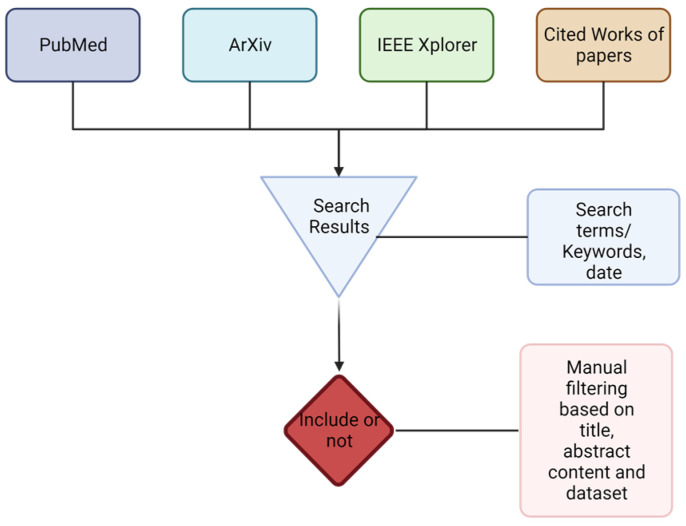
PRISMA-based Flowchart depicting the search and filtering process. Created with BioRender.com.

**Figure 2 diagnostics-13-03140-f002:**
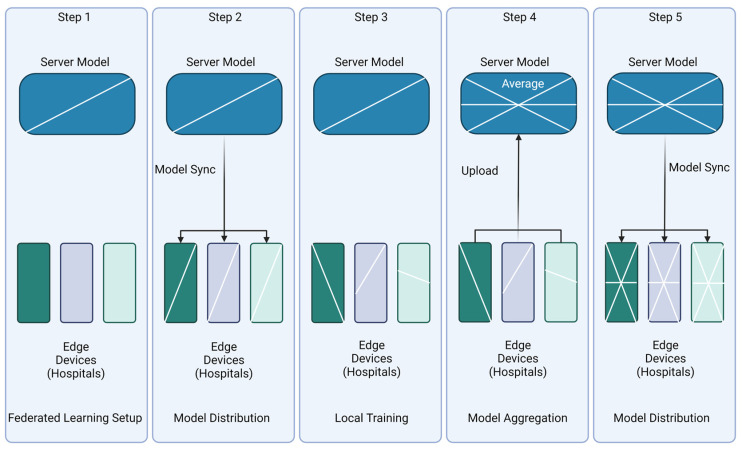
Steps of Federated Learning. The different colors of edge devices represent the client’s setup at various hospitals. The white lines represent a hypothetical skew of each step. Step 1 includes the initialization of the setup. Step 2 involved synchronizing the model by sending the central model from the server to the individual devices to be synced. Step 3 depicts how the bias of the model changes by altering the angle of the white line. Step 4 involves the local models being sent back to the central server to be aggregated. Step 5 depicts the global model then being redistributed to the edge devices. This figure is heavily influenced by [26]. Created with BioRender.com.

**Figure 3 diagnostics-13-03140-f003:**
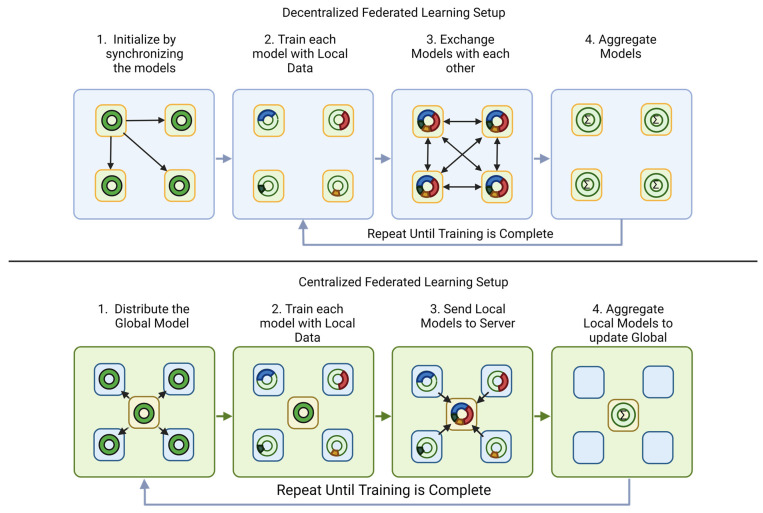
Decentralized (top) vs. Centralized Federated Learning (bottom). The yellow box in the middle of the centralized learning (bottom) represents a central server while the blue represent the individual clients. The green boxes with orange outlines in the de-centralized setup (top) represent the individual clients. Thje dark green circles reprsent the starting model for ech architecture. In step 2 the different color fractions represent the different contribu-tions a local node makes to the overall model. Step 3 shows the transferring of knowledge to the different servers and depicts the stark difference between the two setups. Step 4 depicts how the models aggregate and also depict the differences by which the setups differ. Afterwards the cycle starts over, where that aggregated model then becomes the starting point (dark green circle) for the next iteration Figure heavily influenced by one presented in [27]. Created with BioRender.com.

**Figure 4 diagnostics-13-03140-f004:**
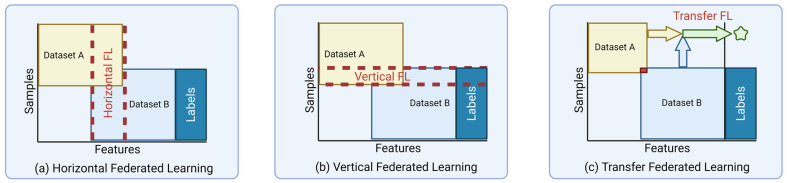
Visual depictions of Horizontal, Vertical, and Transfer Federated Learning. (**a**) During Horizontal FL, the features from one dataset are shared by those of another dataset. (**b**) During Vertical FL, the samples/patients are the same in two datasets, but the features are different. (**c**) During Transfer FL the learnings of one dataset can be applied to another dataset (represented by the star) even though they have little to no overlap of features or samples (depicted by the red box). Figure heavily influenced by thosepresented in. Created with BioRender.com.

**Figure 5 diagnostics-13-03140-f005:**
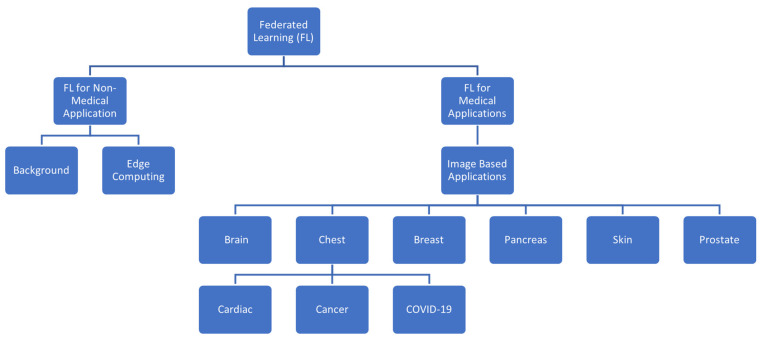
This figure provides a high-level overview of the categories of applications included in this review.

**Figure 6 diagnostics-13-03140-f006:**
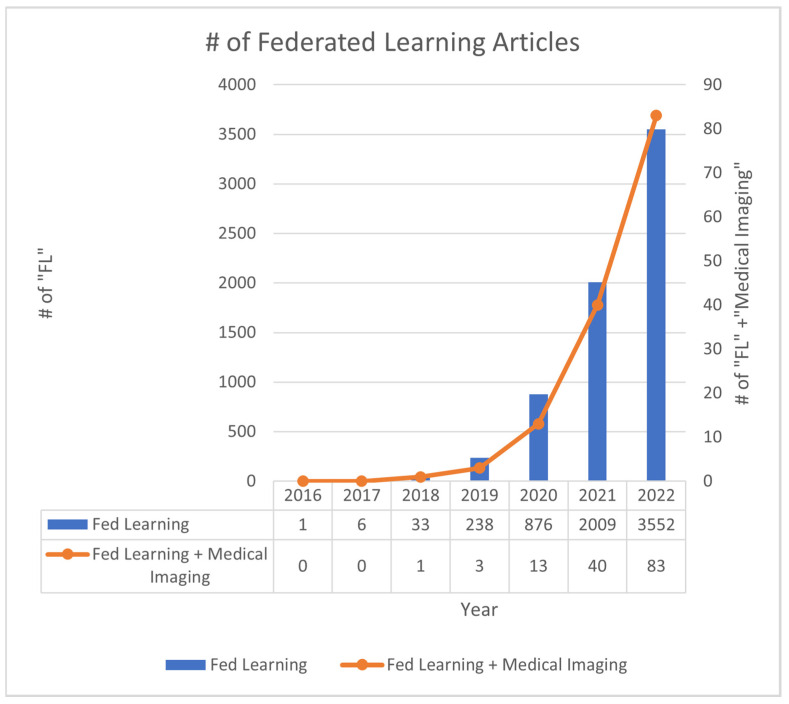
Number (#) of Federated Learning Articles per year. Blue depicts the number of FL articles per year, with the y axis on the left. Orange depicts the number of Medical Imaging related FL articles per year with the y axis on the right. Corporate interests in FL revolve around the ability to gather the learnings of edge devices, such as that on user’s cell phones, without moving large amounts of data that would be otherwise required [1,105,106]. Additionally, the ability for users to maintain their own data governance is also a large selling point, especially when dealing with potentially sensitive information. Larger companies like Google have been utilizing federated learning with its Google keyboard; additionally, IBM has also been utilizing the methods as well [105]. Healthcare companies like Rhino Health utilize FL to create a platform for sharing information amongst institutions.

**Table 1 diagnostics-13-03140-t001:** Literature search results.

Keywords	PubMed	ArXiv	IEEE Xplorer
“Federated Learning”	354	3291	4063
“Federated Learning” + “Medical”	151	59	354
“Federated Learning” + “Healthcare”	90	25	192
“Federated Learning” + “Medical Imaging”	23	41	38
“Federated Learning” + “COVID-19”	43	19	63
“Federated Learning” + “Brain”	18	26	58
“Federated Learning” + “Cancer”	37	5	29
“Federated Learning” + “Breast”	2	3	11
“Federated Learning” + “Pancreas”	1	2	2

**Table 2 diagnostics-13-03140-t002:** Summary of Literature Reviews.

Author	Primary Focus	Specific to Medical Field	Summary/Strengths of the Review
Kamble et al. [13]	Frameworks	Yes	Summarizes applications of FL on medical imaging tasks.
Abreha et al. [14]	Edge Computing	No	Relates FL to Edge computing. Compares methods of learning such as Centralized learning, Deep learning, and Cloud Computing Services.
Aouedi et al. [15]	FL in MedIOT	Yes	Aggregates FL works in MedIoT. Provides information about the variations of FL, such as decentralized vs. centralized as well as the different aggregation techniques. Focuses significantly on COVID-19 applications. Extensive discussion section proposing several future directions.
Beltran et al. [16]	CFL vs. DFL	Yes	Compares and explains the differences between Decentralized FL and Centralized FL. Reviews the applications of DFL and analyzes DFL framework.
Castiglioni et al. [7]	Background of AI in medical imaging	Yes	Provides context surrounding FL. Explains well what AI is and how it is applied to medical images, as well as the challenges at each step.
Chowdhury et al. [2]	Cancer Research FL	Yes	Reviews applications of FL to various forms of cancer.
Crowson et al. [3]	FL in healthcare	Yes	Evaluates the current state of FL in healthcare. Only includes up to 2020, so only 13 sources.
Adamidi et al. [17]	AI in COVID-19	Yes	Conducts a systematic review of published and preprint reports of AI models for Coronavirus disease 2019. Some of the reports include FL applications.
Joshi et al. [8]	FL background and healthcare	Yes	Explains in detail the fundamentals of FL and the possible variations. Introduces various FL applications categorized into prognosis, diagnosis, and clinical workflow.
Mahlool et al. [18]	Applications of FL and DL	Yes	Medical applications of FL and DL.
Kaissis et al. [11]	Security in FL Medical Imaging	Yes	Provides some context surrounding the challenges of security in FL medical imaging. Demonstrates by discussing the different kinds of attacks and the solutions provided by various other works.
Zhang et al. [19]	Security in FL	Yes	Focuses on the challenges of security and proposes novel applications of privacy-preserving FL in the following scenarios: high communication cost, system heterogeneity and statistical heterogeneity. Nicely generalizes how the issues can be fixed.
Li et al. [20]	Applications of FL in Industrial Engineering and healthcare	Some	Discusses the numerous issues that tend to arise when talking about FL. Focuses on the applications related to Industrial Engineering and, secondly, healthcare.
Narmadha et al. [21]	Applications of FL in healthcare	Yes	A high-level review of FL in healthcare
Ng et al. [22]	FL applications with small datasets	Yes	Provides insight into FL in healthcare applications, focusing specifically on how the problem of small datasets can be alleviated through FL and how different applications were trained and implemented. There were only a handful of direct applications. The group highlights four challenges for FL: weight updating, participation incentives, hardware requirement burdens, data heterogeneity and labeling.
Nguyen et al. [10]	Systematic review of FL in healthcare	Yes	Provides insight into some of the other reviews published before this one. Talks about the key principles around FL in healthcare, motivations for using FL in healthcare, requirements for FL and advanced FL designs for healthcare. In Section 5, the paper then goes into the applications of FL in healthcare.
Pfitzner et al. [9]	Extensive review of parameters and application of FL in healthcare	Yes	Extensive systematic review that discusses the concepts and research in FL relevant to healthcare.
Nguyen, T. et al. [12]	FL in Ophthalmology	Yes	FL applications in ophthalmology, as well as some applications on EMR data, Internet of Things in healthcare, as well as medical imaging.
Rauniyar et al. [5]	FL applications in medical field	Yes	Focuses on medical applications rather than technical rigor; provides significant background information as well as information regarding frameworks, challenges, and future directions.
Rootes-Murdy et al. [23]	FL in Neuroimaging	Yes	Provides a summary of federated neuroimaging data analysis tools. The paper also talks about the different platforms available for neuroimaging, such as COINSTAC.
Yang et al. [24]	Technical FL Summary and applications	No	Focuses heavily on the technical aspects and concepts of FL. Provides general applications not specific to the medical field.
Zhou et al. [25]	Review of Deep Learning in medical Imaging	Yes	Focuses on the application of Deep Learning, not specifically FL, in the medical imaging field. Provides insight into the strides that have occurred in various fields, organizing each section by the part of the body.

**Table 3 diagnostics-13-03140-t003:** Summary of Brain-related applications.

Author	Task	Disease	Goal
Sheller et al. [41]	Tumor segmentation	Tumor	Use FL to achieve generalizability of ML models.
Li et al. [43]	Tumor segmentation	Tumor	1. Implement differential privacy and prove feasibility; 2. Test effects of imbalanced training nodes.
Silva et al. [50]	Analysis of subcortical thickness and shape features	Alzheimer’s, Parkinson	Introduce an easy-to-use framework to share any biomedical data with a case study that analyzes subcortical thickness and shape features across diseases such as Alzheimer’s and Parkinson’s, while comparing to healthy individuals.
Roy et al. [28]	Whole brain segmentation	General	Create a central server-less FL system.
Sheller et al. [42]	Tumor segmentation	Tumor	Use FL to increase 1. Generalizability; 2. Performance.
Silva et al. [51]	Analysis of subcortical thickness and shape features	Alzheimer’s, Parkinson	A case study that analyzes subcortical thickness and shape features across diseases such as Alzheimer’s and Parkinson’s while comparing them to healthy individuals.
Stripelis et al. [39]	Brain Age prediction	Dementia	Demonstrate an approach to address heterogeneous environments by predicting Dementia using Brain Age.
Stripelis et al. [31]	Brain Age prediction	Dementia	1. Demonstrate a successful implementation of Cheon-Kim-Kim-Song scheme for a more secure Transfer supporting fully homomorphic encryption; 2. Demonstrate performance on skewed data.
Li et al. [35]	Autism spectrum disorder biomarker discovery	Autism	1. Privacy-preserving pipeline for fMRI; 2. Address data heterogeneity due to domain shift.
Huang et al. [52]	Detection and stage classification of Alzheimer’s	Alzheimer’s	1. Set up a way to conduct multisite Alzheimer’s classification by 3D convolutional neural network and t1w MRI; 2. Compare results to other models.
Bercea et al. [55]	Brain anomaly segmentation	General	Create a framework that can identify anomalies by only sending shape and intensity parameters.
Machler et al. [45]	Tumor segmentation	Tumor	Create a better way to average updated model weights.
Fan et al. [58]	Autism spectrum disorder diagnosis	Autism	1. Create an FL framework for analyzing 3D Brain MRI images; 2. Implement privacy measures to enhance security.
Parekh et al. [56]	Organ localizing, lesion segmentation	General	1. Demonstrate the feasibility of training cross-domain; 2. cross-task FL models.
He et al. [46]	Image classification	Tumor	Implement a simple cosine-based nonlinear quantization to achieve results in compressing round-trip communication costs.
Dipro et al. [54]	Image classification	Parkinson’s	A novel approach to detecting Parkinson’s disease with FL.
Zhang et al. [29]	Tumor segmentation	Tumor	Create a new FL method to overcome the performance drops from data heterogeneity.
Liu et al. [60]	Lesion segmentation	Multiple Sclerosis	Create a framework that addresses domain shifts that are specific to Multiple Sclerosis lesion segmentation tasks.
Stripelis et al. [53]	Brain classification	Alzheimer’s and Brain Age	Build an architecture 1. That encrypts parameters before transmission, computes models via homomorphic encryption and uses methods to limit leakage; 2. Performs well across heterogeneous environments.
Islam et al. [48]	Image classification	Tumor	First study to use Complex CNN model for FL MRI-based tumor classification.
Huang et al. [61]	Metastasis Segmentation	Brain Metastasis	Overcome catastrophic forgetting by implementing Continual Learning on Brain Metastasis Identification.
Zeng et al. [62]	Image classification	Schizophrenia, Major Depressive Disorder	Propose a 2-stage method of gradient matching that aims to reduce domain discrepancy. The group demonstrated the ability of this method on resting-state functional MRIs for diagnostic classification.
Ads et al. [34]	Image classification	Tumor	Implement both split learning and Vertical distribution for brain tumor classification.
Elmas et al. [64]	MRI reconstruction (Not Diagnostic)	General	Introduce FedGIMP for MRI reconstruction, which leverages a 2-stage approach: cross-site learning of generative MRI prior and prior adaption following injection of the imaging operator.
Fay et al. [44]	Tumor segmentation	Tumor	Implement a Private Aggregation of Teacher Ensembles based on the FL model on the BraTS dataset.
Guo et al. [63]	MRI reconstruction (Not Diagnostic)	General	1. Introduce a method called FL-M that enables multi-institutional collaborations for MRI reconstruction; 2. Address domain shift issues by aligning the latent space distribution between the source and target domain; 3. Conduct experiments that provide insights about FL in MRI reconstruction.
Gupta et al. [57]	Brain Age prediction	General	Demonstrate the ability to conduct membership interference attacks on deep learning models.
Pati et al. [49]	Tumor segmentation	Tumor	Conduct experiments on the largest dataset to date regarding the feasibility and effects of FL on glioblastoma sub-compartment boundary detection.
Shamseddine et al. [59]	Autism spectrum disorder diagnosis	Autism	Use FL models to determine if a patient has Autism or not based on: 1. behavioral screening data; 2. A clear facial picture.
Rawat et al. [47]	Tumor segmentation	Tumor	Introduce robust learning protocol, which is a combination of server-side adaptive optimization and parameter aggregation schemes to tackle data heterogeneity issues and communication cost of training.
Knolle et al. [36]	Pancreas segmentation and tumor segmentation	General pancreas and tumor	Create an FL architecture that can operate in resource-constrained environments by decreasing the amount of image features being used and transferred.

**Table 4 diagnostics-13-03140-t004:** COVID-19 Applications.

Author	Task	Modality	Goal
Liu et al. [65]	Classification	CXR	Compare distributed learning/FL to four other classic models.
Xu et al. [66]	Classification	CT	Introduce UCADI, a global AI CT model collaborative; Create a large COVID-19 dataset;
Kumar et al. [67]	Segmentation/Classification	CT	Create a new data normalization technique; COVID-19 detection technique; introduce blockchain. And create a new dataset.
Lydia et al. [68]	Classification	CXR	Create an FL-based COVID-19 detection model on an Internet of Things, enabling edge computing environment.
Dayan et al. [69]	Classification	CXR + EMR	Provide proof of concept that will demonstrate the ability to create an FL model that can be used across heterogeneous, unharmonized datasets for the prediction of clinical outcomes in patients with COVID-19.
Zhang et al. [70]	Classification	CT, CXR	Create a more communication-efficient FL technique that also better handles data heterogeneity;
Dou et al. [71]	Segmentation/Classification	CT	Demonstrate the feasibility of FL for detecting COVID-19-related CT abnormalities in a multinational study; Longitudinal case studies to estimate lesion burden;
Feki et al. [72]	Classification	CXR	Demonstrate the ability to implement FL on COVID-19 detection in X-rays; Demonstrate the ability of FL to overcome non-IID issues;
Yang et al. [37]	Segmentation/Classification	CT	Use FL on chest CT to demonstrate COVID-19 detection capabilities in a multinational study that could demonstrate the ability to overcome domain shift; Implement semi-supervision to help increase annotated dataset size to improve FL performance;
Salam et al. [73]	Classification	CXR + EMR	Prove Efficacy of FL vs. traditional learning in CXR COVID-19 detection; Determine which parameters affect prediction accuracy;
Alam et al. [74]	Segmentation/Classification	CXR	Explore 2 decision-making tasks, COVID-19 detection and lung area segmentation detection of chest radiology images; Compare the abilities of a high-end computer and a low-end computer; the low-end did better on lung segmentation and high-end did better on COVID-19 detection;
Liang et al. [75]	Segmentation/Classification	CT+ EMR	Create a framework where CT + EMR information can be used to diagnose COVID-19; Create a model that can automatically segment lung lesions to keep track of progress;
Zhang et al. [30]	Segmentation	CXR	Create a privacy-preserving data augmentation method enhancing security;
Ho et al. [76]	Classification	CXR + EMR.,	Construct an FL system using chest X-rays and symptom information; Add spatial pyramid pooling to a 2D convolutional neural network to improve accuracy; Explore how different parameters can improve accuracy for non-IID data; Apply a differential privacy stochastic gradient descent to improve the privacy of patient data;
Qayyum et al. [77]	Classification	CXR + ultrasound	1. Create a clustered FL method to develop a multimodal COVID-19 FL detection system using X-ray and ultrasound;
Durga et al. [78]	Classification	CT	1. Propose a novel framework based on blockchain and FL model;
Zheng Li et al. [79]	Classification	CXR	1. Create a FL framework with a dynamic focus on COVID-19 detection on CXR;

**Table 5 diagnostics-13-03140-t005:** Applications of FL to Pancreatic Imaging.

Author	Task	Goal
Wang et al. [81]	Pancreas segmentation	Generate and evaluate an FL model for pancreas segmentation.
Shen et al. [82]	Pancreas segmentation	Investigate heterogeneous optimization methods that show improvements for the automated segmentation of pancreas and pancreatic tumors in abdominal CT images.
Knolle et al. [36]	Pancreas segmentation	Create an FL architecture that can operate in resource-constrained environments by decreasing the amount of image features being used and transferred.

**Table 7 diagnostics-13-03140-t007:** Applications of FL to Skin Diseases.

Author	Task	Disease	Goal
Hashmani et al. [86]	Segmentation and classification	Skin tumor	Propose an adaptive FL-based skin disease model to create an intelligent dermoscopy device.
Mou et al. [87]	Segmentation and classification	Melanoma detection	Present a feasibility study that demonstrated the capabilities of FL on medical records.
Hossen et al. [88]	Classification	Skin diseases	1. Create a custom image dataset prepared with 4 distinct classes of skin disease; 2. Create a novel CNN model to classify the four disease types; 3. Use FL to enhance the security of medical imaging using the custom dataset.
Wicaksana et al. [89]	Classification	Skin lesions	Introduce and implement CusFL, a method in which each client trains a private model based on the global model aggregated from all private models trained in the immediate previous iterations.

**Table 8 diagnostics-13-03140-t008:** Applications of FL to Prostate Imaging.

Author	Task	Goal
Yan et al. [91]	Prostate classification	Introduce the VAFL framework; Improve the performance of the global model for classification; Reduce variation while not increasing communication burden; limit the amount of training data.
Wicaksana et al. [89]	Prostate classification	Introduce and Implement CusFL, a method in which each client trains a private model based on the global model aggregated from all private models trained in the immediate previous iterations.
Sarma et al. [92]	Prostate segmentation	Demonstrate the ability to train a FL model across 3 academic institutions while preserving patient privacy.

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
