# Peer review of "Medical Imaging Applications of Federated Learning"

_diagnostics, 2023, doi:10.3390/diagnostics13193140_

Round 1

Reviewer 1 Report

·        In line 17 “ML model” is just an abbreviation. It is appropriate to shorten it by giving the long version.

·        In section 3. Results: It contains examples that give both the objectives of the articles and the selection of the articles. Section 4 is also similar. However, the detail of article selection has been explained in great detail; in fact, explanations can be made through Table 1. 

·        In Section 4, it is defined by only one table. The title "4. Background" should be rewritten, including subheadings.

·        In Table 2 is very difficult to understand and follow in terms of form. In Table 2 takes up many pages, maybe it could be designed horizontally. The same thing can be applied to other tables.

·        In line 749: “Coporate intersts inf FL revolve around the ablity to gather the learnings of edge devicessuch as that on user’s cell phones without moving large amounts of data that would be otherwise required.” Sentence a should be corrected to take into account spelling errors. And the sentence should be cited.

No comment

Author Response

Comments 1: In line 17 “ML model” is just an abbreviation. It is appropriate to shorten it by giving the long version.

Response 1: Thank you for pointing this out. This has been reviewed and edited.

Comments 2: In section 3. Results: It contains examples that give both the objectives of the articles and the selection of the articles. Section 4 is also similar. However, the detail of article selection has been explained in great detail; in fact, explanations can be made through Table 1.

Response 2: Thank you for you comment, I have addressed it by removing a few details, summarizing others and directly referencing the table.

Comments 3: In Section 4, it is defined by only one table. The title "4. Background" should be rewritten, including subheadings.

Response 3: Thank you for your comment, I have relocated one of the paragraphs from the results section to the background section to preface the table better. Additionally, I have fixed the heading.

Comments 4: In Table 2 is very difficult to understand and follow in terms of form. In Table 2 takes up many pages, maybe it could be designed horizontally. The same thing can be applied to other tables.

Response 4: Thank you for your comments and suggestions. I have included the revised (landscape) tables in place of the original tables. 

Comments 5: In line 749: “Coporate intersts inf FL revolve around the ablity to gather the learnings of edge devicessuch as that on user’s cell phones without moving large amounts of data that would be otherwise required.” Sentence a should be corrected to take into account spelling errors. And the sentence should be cited.

Response 5: Thank you for your fine comment. I have revised this sentence and added citations 4,105 and 106.

Reviewer 2 Report

The provided review report meets the standards of Diagnostic Journal and can be considered for publishing in this journal. Please recheck some the references (such as No., 52) to revise them according to the pre-defined structure of the journal. Since, some of medical images are taken for non-diagnostic aims (such as patient setup or tumor tracking at Radiotherapy), please indicate that the focus of this work is on medical images with diagnostic aims. Please re-check the English of the work asking natives.

The provided review report meets the standards of Diagnostic Journal and can be considered for publishing in this journal. Please recheck some the references (such as No., 52) to revise them according to the pre-defined structure of the journal. Since, some of medical images are taken for non-diagnostic aims (such as patient setup or tumor tracking at Radiotherapy), please indicate that the focus of this work is on medical images with diagnostic aims. Please re-check the English of the work asking natives.

Author Response

Comments 1: The provided review report meets the standards of Diagnostic Journal and can be considered for publishing in this journal. Please recheck some the references (such as No., 52) to revise them according to the pre-defined structure of the journal.

Response 1: Thank you for pointing this out. I agree with this comment. Therefore, I have made the correct revisions by making sure the references follow the journal’s format.

Comments 2: Since, some of medical images are taken for non-diagnostic aims (such as patient setup or tumor tracking at Radiotherapy), please indicate that the focus of this work is on medical images with diagnostic aims.

Response 2: Thank you for pointing this out. I have, accordingly, added in the corresponding tables as well as in the body when there are instances where the work does not focus on diagnostic goals when using medical imaging. Please refer to Table 3, the last sentence of section 5.1.4.6, and the second paragraph of Section 5.7.

4. Response to Comments on the Quality of English Language

Point 1: Please re-check the English of the work asking natives.

Response 1:   Thank you for pointing out the need for further work. The paper has been reread and reworded in a few areas.

Reviewer 3 Report

This is a systematic review of Federated Learning in medical imaging. It follows the PRISMA guidelines and is well-written.

I only have one comment. There is no mention of recording of video data during procedures such as laparoscopic procedures and robotic-assisted surgeries. Are there any issues that apply to this type of data that differ from the data that is mentioned. As opposed to the field of Radiomics, Surgomics applies to this type of data.

Several articles are noted below for your reference.

Wagner M, Brandenburg JM, Bodenstedt S, Schulze A, Jenke AC, Stern A, Daum MTJ, Mündermann L, Kolbinger FR, Bhasker N, Schneider G, Krause-Jüttler G, Alwanni H, Fritz-Kebede F, Burgert O, Wilhelm D, Fallert J, Nickel F, Maier-Hein L, Dugas M, Distler M, Weitz J, Müller-Stich BP, Speidel S. Surgomics: personalized prediction of morbidity, mortality and long-term outcome in surgery using machine learning on multimodal data. Surg Endosc. 2022 Nov;36(11):8568-8591. doi: 10.1007/s00464-022-09611-1. Epub 2022 Sep 28. PMID: 36171451; PMCID: PMC9613751.

Gumbs AA, Croner R, Abu-Hilal M, Bannone E, Ishizawa T, Spolverato G, Frigerio I, Siriwardena A, Messaoudi N. Surgomics and the Artificial intelligence, Radiomics, Genomics, Oncopathomics and Surgomics (AiRGOS) Project. Art Int Surg 2023;3:180-5. http://dx.doi.org/10.20517/ais.2023.24

Silas MR, Grassia P, Langerman A. Video recording of the operating room--is anonymity possible? J Surg Res. 2015 Aug;197(2):272-6. doi: 10.1016/j.jss.2015.03.097. Epub 2015 Apr 9. PMID: 25972314.

Gordon L, Reed C, Sorensen JL, Schulthess P, Strandbygaard J, Mcloone M, Grantcharov T, Shore EM. Perceptions of safety culture and recording in the operating room: understanding barriers to video data capture. Surg Endosc. 2022 Jun;36(6):3789-3797. doi: 10.1007/s00464-021-08695-5. Epub 2021 Oct 4. PMID: 34608519.

Prigoff JG, Sherwin M, Divino CM. Ethical Recommendations for Video Recording in the Operating Room. Ann Surg. 2016 Jul;264(1):34-5. doi: 10.1097/SLA.0000000000001652. PMID: 27123809.

Author Response

Comments 1: I only have one comment. There is no mention of recording of video data during procedures such as laparoscopic procedures and robotic-assisted surgeries. Are there any issues that apply to this type of data that differ from the data that is mentioned. As opposed to the field of Radiomics, Surgomics applies to this type of data.

Several articles are noted below for your reference.

Response 1: Thank you for your comments. There is a very recent article from June 2023, that did in fact use Federated Learning during laparoscopic procedures to recognize surgical phase. I have added it as reference # 104.

H. Kassem, D. Alapatt, P. Mascagni, A. Karargyris and N. Padoy, "Federated Cycling (FedCy): Semi-Supervised Federated Learning of Surgical Phases," in IEEE Transactions on Medical Imaging, vol. 42, no. 7, pp. 1920-1931, July 2023, doi: 10.1109/TMI.2022.3222126.

Additionally, The following sentence was added to the paper:

“A noteworthy variation was conduction by Kassem et al. who use surgical videos rather than images to apply FL for surgical phase detection (Not diagnosis) on Cholecystectomy procedures [104].”
